# In Vitro and In Silico Antidiabetic Efficacy of Solanum lasiocarpum Dunal Fruit Extract

Jing Zhao[1◦], Ahmed Abdulkareem Najm[1◦], Ibrahim Mahmood[3], Zhang Yu Ming[2], Partha Pratim Dutta[4], Wamidh H. Talib [5], Douglas Law [2,6]*, Shazrul Fazry [1,7]*

1 Department of Food Science, Faculty of Science and Technology, Universiti Kebangsaan Malaysia, Bangi, Selangor, Malaysia, 2 Faculty of Health and Life Sciences, Inti International University, Nilai, Negeri Sembilan, Malaysia, 3 Dentistry Department, Al-Rafidain University College, Baghdad, Iraq, 4 Faculty of Pharmaceutical Science, Assam down town University, Panikhaiti, Guwahati, Assam, India, 5 Faculty of Allied Medical Sciences, Applied Science Private University, Amman, Jordan, 6 ENIAC, Centro Universitario de Excelencia. R. Força Pública, 89, Centro Guarulhos-S.P., Brazil, 7 Tasik Chini Research Center, The Centre for Natural and Physical Laboratory Management, Universiti Kebangsaan Malaysia, Bangi, Selangor, Malaysia,

◦ These authors contributed equally to this work.
* shazrul@ukm.edu.my (SF); douglas.law@newinti.edu.my (DL)

## Abstract

Exploring the possibility of familiar dietary sources as additional diabetes treatments is crucial, especially considering the financial difficulties related to diabetes mellitus. Using both in vitro and in silico techniques, this work aims to assess the antidiabetic benefits of extract from *Solanum lasiocarpum Dunal*. The evaluations encompass the ability to scavenge DPPH radicals, inhibition of α-amylase, α-glucosidase, inhibition of DPP-4, cytotoxicity, and glucose absorption kinetics. With an IC50 value of 0.69 ± 0.14 mg/ml, *S. lasiocarpum* showed encouraging DPPH inhibition. IC50 values of 2.123 ± 0.14 mg/ml inhibited the enzymes α-amylase, α-glucosidase, and DPP-4. Furthermore, a notable increase (P < 0.05) in glucose uptake by L6 myoblasts was observed with the administration of various combinations. In silico analysis, including XP docking and MM-GBSA, revealed that 10 and 21 compounds within the combination exhibited substantial interactions and stable binding capabilities with α-amylase and DPP-4 proteins, indicating their potential as enzyme inhibitors. Therefore, it can be inferred that *S. lasiocarpum* represents a promising therapeutic approach for diabetes management.

## Introduction

Diabetes mellitus (DM) is a common metabolic disease marked by resistance to insulin or ineffective synthesis of insulin, which results in noticeably high blood glucose levels [1]. Typical symptoms include increased thirst, frequent urination, impaired eyesight, and weight loss [2]. The prevalence of diabetes has increased dramatically worldwide, putting a significant financial strain on countries and communities. Currently, insulin and oral hypoglycemic medications are the leading therapies for diabetes. However, these drugs frequently have side effects and lead to drug resistance, which makes people look for safer and more effective substitutes [3].

**Data availability statement:** All relevant data are within the manuscript and its Supporting Information files.

**Funding:** This work was supported by the Department of Food Sciences, Faculty of Science and Technology, Universiti Kebangsaan Malaysia (Funding No.: ST-2021-010) Role: publication fee, Data collection, and data analysis. Inti International University Seed Grant Funding (INTI-FHLS-01-03-2022, INTI-FHLS-11-03-2022 and INTI-FHLS-01-26-2023) Role: Funding the publication fee (partially).

**Competing interests:** The authors have declared that no competing interests exist.

**Abbreviations:** EPOCH, Not explicitly mentioned in the context; ADMET, Absorption, Distribution, Metabolism, Excretion, and Toxicity; Lipinski's five principles, A set of rules for drug-like properties; MM-GBSA, Molecular Mechanics-Generalized Born Surface Area; XP GScore, A method for predicting the solubility of small molecules; QikProp, A Schrödinger's software suite module for predicting various drug-like properties; logS, Logarithm of the solubility of a compound in water; QPP Caco, A value used to predict the permeability of a compound across the Caco-2 cell monolayer; DM, Diabetes Mellitus; DMEM, Dulbecco's Modified Eagle Medium; DMSO, Dimethyl Sulfoxide; DPPH, 2, 2-diphenyl-1-picrylhydrazyl; DPP-4, Dipeptidyl Peptidase 4; FBS, Fetal Bovine Serum; LC-MS, Liquid Chromatography Mass Spectrometry; PBS, Phosphate Buffered Saline; SRB, Sulphorhodamine B; T1DM, Type 1 Diabetes Mellitus; T2DM, Type 2 Diabetes Mellitus; TFC, Total Flavonoid Content; TPC, Total Phenolic Content; WHO, World Health Organization.

In the study and creation of pharmaceuticals, herbal extracts are a crucial source of supplemental and therapeutic pharmacologically active compounds [4,5]. Compared to synthetic medications, they are considered safer, more readily available, affordable, and more potent [6]. Historically, herbal remedies have been used to treat diabetes in many nations [7]. In addition to other mechanisms of action, herbal extracts inhibit α-amylase and α-glucosidase, enzymes responsible for breaking down carbohydrates into glucose. This inhibition results in a slower release of glucose into the bloodstream, thereby helping in better glycemic control [8].

In Malaysia, *Solanum lasiocarpum* Dunal (Solanaceae) is known as Terung Dayak or Terung Asam and is particularly prevalent in Sarawak and Sabah. It is often used in local cuisine for its sour flavour, added to curries and sauces. In India, *S. lasiocarpum* is found in the North Eastern states, particularly in regions like Assam. The fruit is also used in local dishes for its sour taste. In traditional medicine, various plant parts are employed to treat ailments such as fever, cough, and toothache. *S. lasiocarpum,* native to Southern China, particularly the Guangxi and Hunan provinces, has been used for over three centuries as a medicinal and edible plant [9,10]. Recent research suggests that mogrosides and polysaccharides extracted from *S. lasiocarpum* are effective in regulating insulin resistance, thereby contributing to antidiabetic effects and blood sugar level control [11,12]. Clinical trials and further research are necessary to validate its efficacy and safety in broader populations, but current evidence suggests a promising future for *S. lasiocarpum* in diabetes management [9]. In Malaysia, the plant is commonly referred to as "terung asam" and is primarily used for its medicinal properties. A decoction of its roots alleviates violent body pains and discomfort after meals. The roots are also used in baths to treat fever and in poultices for treating itches, cuts, wounds, and severe bruises. In India, *S. lasiocarpum* is utilised similarly for its therapeutic properties. It is applied in traditional medicine for treating conditions like syphilis and to reduce swelling by using the leaves as poultices [13–15]. In traditional medicine, *S. lasiocarpum* has been utilised for its antidiabetic properties. According to Li [16], the fruit extract is typically used in doses ranging from 50 to 200 mg/kg for various ailments. Previous studies, such as those by Nguyen et al. [17], have demonstrated significant antidiabetic effects at 100 mg/kg doses in streptozotocin-induced diabetic rats. WHO monographs [18] further support its medicinal potential, detailing its traditional uses and effective dose ranges. These sources provide a comprehensive understanding of the traditional and scientifically validated doses of *S. lasiocarpum*, highlighting its relevance and efficacy in conventional medicine. Therefore, this study aims to investigate the antidiabetic effects of *S. lasiocarpum* through both *in vitro* and *in silico* methods.

## Materials & methods

**Chemicals and reagents.** Several chemicals and reagents were procured from various suppliers for this study. Sigma-Aldrich (USA) PNPG, iodine, Trolox, α-glycosidase, α-amylase, sodium hydroxide (NaOH), sodium chloride (NaCl), quercetin, gallic acid, sodium nitrite (NaNO2), and DPPH (2,2-diphenyl-1-picrylhydrazyl) were supplied. Merck (USA) supplied starch, Folin-Ciocalteu reagent, and Aluminum chloride solution (AlCl$_3$). Fisher Scientific (USA) provided DMEM (Dulbecco's Modified Eagle Medium), fetal bovine serum (FBS), ethanol, methanol, and hydrochloric acid (HCl). All chemicals and reagents used were of analytical grade.

**Preparation of extract.** Mature fruits of *S. lasiocarpum* from Sarawak, Malaysia, were dried at 40°C in an oven. After removing the outer layer, the fruits were ground into a powder using an electric grinder. This powder was mixed with water in a 1:10 ratio (weight/volume), thoroughly blended, and subjected to reflux extraction using an ultrasonic device at 60°C.

The mixture was then filtered and concentrated under reduced pressure to produce the crude aqueous extract of *S. lasiocarpum*.

**Quantification of flavonoids and phenolic compounds.** To measure the Total Phenolic Content (TPC), we followed the method described by Shahid et al. [19]. A 96-well plate was filled with 10 µL of the extract (1 mg/mL), 20 µL of Folin-Ciocalteu (FC) reagent, and 70 µL of deionised water. Following a 5-minute incubation period, 100 µL of a 20% NaCO3 solution was added, and the reaction continued in the absence of light for an additional half-hour. At 730 nm, absorbance was determined with a microplate reader (Tecan, Switzerland). Gallic acid equivalents (GAE) per gram of extract were used to compute TPC. We employed Salahuddin et al.'s technique for Total Flavonoid Content [20]. In a 96-well plate, 30 µL of extract (1 mg/mL) was combined with 120 µL of distilled water and 9 µL of 5% NaNO2 solution. Nine µl of a 10% AlCl3 solution was added after five minutes, and the reaction continued for another 5 minutes. Subsequently, 72 µL of distilled water and 60 µL of 1 M NaOH were added, and absorbance was measured at 510 nm. TFC was expressed as milligrams of quercetin equivalents (QE) per gram of extract.

**Assessment of DPPH radical scavenging activity.** Different volume samples of *S. lasiocarpum* aqueous extracts were made in the range (0 to 5 mg/ml). 50 µL of the sample solution and 150 µL of the DPPH solution were added to each 96-well plate. For thirty minutes, the plate was incubated in the dark. Absorbance was determined at 517 nm using a microplate reader by Dappula et al. [21]. DPPH radical scavenging activity was calculated using the formula:

$$Scavenging\ activity\ = \left(Absorbance(b) - Absorbance(s)\,/\,Absorbance(b)\right) \times 100\%$$

The term " $Absorbance(b)$ " denotes the absorbance of the blank, whereas " $Absorbance(s)$ " refers to the absorbance of the sample.

## *In vitro* enzyme inhibition assays

**α-Amylase inhibition assay.** The iodine starch method, as modified by Perumal et al. [22], was used to evaluate the inhibition of α-amylase. 50 µL of crude extract was mixed with 60 µL of 0.1 M PBS (pH 6.9), 20 µL of α-amylase (1U/mL), and incubated at 37°C for 30 minutes. Then, 50 µL of 0.25% starch solution was added and incubated for another 15 minutes. The reaction was stopped with 20 µL of 1 M HCl, followed by heating at 75°C for 5 minutes. After cooling, 50 µL of iodine reagent was added, and absorbance was measured.

**α-Glucosidase inhibition assay.** A modified version of the technique described by Rakariyatham et al. [23] assessed α-glucosidase inhibition. Samples were divided into six gradients of concentration. A 30-minute reaction took place at 37°C with 10 µL of the diluted sample, 130 µL of 50 mM phosphate-buffered saline (pH 6.5), and 10 µL of α-glucosidase (1.5 U/mL) combined. After adding 50 µL of 1 mM PNPG solution and additional incubation, 50 µL of 0.2 M Na2CO3 solution was added to terminate the reaction. Absorbance was measured at 405 nm. DMSO was used as the control, with acarbose as the positive control.

**DPP-4 Inhibition assay.** Chaipoot et al. [24] measured the DPP-4 inhibitory activity using a DPP-4 inhibitor screening test kit (MAK203, Sigma-Aldrich, Germany). The reaction mixtures were incubated for 10 minutes at 37°C and contained 49 µL of DPP4 assay buffer, 1 µL of DPP4 enzyme, 25 µL of the sample, and 50 µL of the inhibitory reaction mix. In kinetic mode, the fluorescence was seen at 37°C for 15 to 30 minutes using a microplate reader.

**Cytotoxic activity on cell lines Cell culture** L6 rat skeletal muscle (myoblast) cells were cultured according to Liu et al. [25]. Cells were obtained from the American Type Culture Collection (ATCC) and cultured in DMEM supplemented with 10% FBS, 1% penicillin, and 1% streptomycin in a CO2-regulated incubator at 37°C.

**Cell cytotoxicity using SRB assay**   The SRB (Sulforhodamine B) assay, as described by Azfaralariff et al. [26], was used to evaluate the cytotoxicity of *S. lasiocarpum*. L6 cells were seeded at a density of $1 \times 10^4$ per well in a 96-well plate, differentiated into myotubes over seven days with 2% DMEM, and treated with different concentrations (0 to 5 mg/mL) of *S. lasiocarpum* for 24, 48, and 72 hours. Cells were fixed with 50% TCA, stained with 0.4% SRB in 1% acetic acid, and absorbance was measured at 510 nm using a BioTek Instruments EPOCH plate reader.

## Glucose absorption assay using 2-NBDG

Following the protocol from a commercial glucose uptake assay kit (KA4086, Abnova), L6 cells were seeded on 96-well plates and differentiated for up to 7 days. Cells were incubated overnight in a serum-free medium, rinsed with KRPH (Krebs-Henseleit) buffer, treated with or without insulin, and exposed to 2-NBDG. Absorbance was measured at 570/610 nm using a BioTek Instruments EPOCH plate reader.

## Kinetic study of 2-NBDG uptake

L6 myoblast cells were seeded into 96-well plates and differentiated into myotubes. Cells were treated with crude extract, incubated with 2-NBDG, and analysed at various intervals using a Thermo Fisher Scientific Varioskan Flash spectral scanning multimode scanner. This methodology was adapted and modified based on the work of Zou et al. [27].

## DPP-4 Inhibition assay

The DPP-4 inhibitory activity of *S. lasiocarpum* was assessed using the DPP-4 inhibitor screening kit (KA 1311; Abnova). The fluorogenic substrate Gly-Pro-AMC was used to measure activity, and absorbance was measured using the Varioskan Flash spectral scanning multimode scanner.

## LCMS analysis

In this study, running in-vitro experiments before in-silico simulations helps validate hypotheses, provides tangible results, and offers a realistic environment to study biological interactions, which in-silico models may not fully capture. Additionally, in-vitro experiments provide insights into biological mechanisms like active-compound interactions. Using an SB-C18 column (150 mm × 4.6 mm × 5 µl), the Agilent 1260 UPLC/ 6540 Q-TOF was used to examine the chemicals present in the extracts. The solvent gradient comprised methanol (solvent A) and water (solvent B). The column was kept at 30°C while the sample was eluted into the electrospray ionisation (ESI) mode at a mobile phase flow rate of 1 ml/min. The gradient elution process for the ionisation mode is as follows: 0–25 min, 20%–100% solvent A; 25–30 min, 100%–0% solvent A.

## *In silico* studies

**Ligand and protein preprocessing.**  Active ingredients identified by LCMS were obtained from PubChem and processed using Schrodinger's LigPrep module. Crystal structures of DPP-4 (PDB ID: 6B1E Chain A) and α-amylase (PDB ID: 3DHP) were retrieved from RCSB PDB and prepared using Schrodinger's Protein Preparation Wizard.

**Molecular XP docking and MM-GBSA calculation.**  Molecular docking was performed using Schrodinger Maestro 13.5. The binding site of the α-amylase protein was predicted using Schrodinger's SiteMap module, and active sites were identified using the Receptor Grid Generation module. Ligands were docked using Glide XP.

**Lipinski parameters and ADMET analysis.** The ADME/T properties, encompassing absorption, distribution, metabolism, excretion, and toxicity, were evaluated using the QikProp module in the Schrodinger software suite described by Hussain et al. [28]. Parameters assessed included hydrophobicity (QPlogPo/w), permeability (QPPCaco), human oral absorption percentage, molecular weight (MW), solubility (QPlogS), and the total solvent-accessible surface area (SASA), among others. Following this, the compounds were filtered based on Lipinski's Rule of Five criteria, following the guidelines outlined by Hajji et al. [29].

**Molecular dynamics simulations.** Using the Desmond program in Schrodinger Maestro 13.5, molecular dynamics simulations were carried out to fine-tune the binding mechanisms of the protein-peptide complexes. The protein and tiny molecules were parameterised using the opls2005 force field, while the water solvent was modelled using the TIP3P model. To preserve electrical neutrality, sodium and chloride ions were supplied at a concentration of 0.150 M while the protein-small molecule combination was submerged in a cubic water volume. Initially, the steepest descent approach, which took over 50,000 steps, decreased the system's energy. Then, during the Network Virtual Terminal and the amount of substance (N), pressure (P), and temperature (T) are conserved (NPT) equilibration phases—each needing an extra 50,000 steps—the placements of the heavy atoms were limited. These phases stabilised the system at a pressure of 1 bar and a temperature of 300 K. After equilibration, a 100 ns simulation was conducted without any constraints. Dynamic trajectory animations and interaction analyses were performed using Schrodinger Maestro 13.5.

## Statistical analysis

All spectrophotometric experiments were conducted in triplicate. Data were presented as mean values with the standard error of the mean (SEM) when appropriate. A regression curve was plotted to determine the precise 50% inhibitory concentration ($IC_{50}$) by graphing the inhibitory percentage of each sample against its concentration. The data were analysed using a one-way ANOVA with a significance level of $p < 0.05$, followed by Tukey's HSD post hoc test. Statistical analyses were performed using SPSS version [25].

## Results

### Quantification of overall phenolic content (TPC) and flavonoid content (TFC)

TPC of each sample was determined using the Folin-Ciocalteu method, while TFC was measured using the $AlCl_3$ Colorimetric method. Standard curve formulations for quercetin and gallic acid were employed in the quantification process. The sample, S. lasiocarpum, showed the highest TPC and TFC values compared to other ratios. As per the analysis, the TPC of S. lasiocarpum was 6.4 ± 0.72 mg GAE/g, while TFC was 1.689 ± 0.17 mg QE/g, as detailed in Table 1.

### Assessment of DPPH radical scavenging activity

A popular approach for assessing the antioxidant activity of food and plant extracts is the DPPH radical scavenging activity assay. According to Alwasel and Gulcin [30], this approach is appreciated for its practicality, simplicity, sensitivity, rapidity, and repeatability. Fig 1 illustrates *S. lasiocarpum's* ability to scavenge DPPH radicals. With an $IC_{50}$ of 10.2 ± 0.11 mg/mL, *S. lasiocarpum* demonstrated DPPH scavenging activity. However, it was not as effective as Trolox, which had an IC50 of 0.69 ± 0.14 mg/mL.

**Table 1. Total phenolic content and total flavonoid content.**

| Sample | Total flavonoid content (TFC) (mg QE/g) | Total phenolic content (TPC) (mg GAE/g) |
|---|---|---|
| *S. lasiocarpum* | 1.689 ± 0.17[a] | 6.4 ± 0.72[a] |

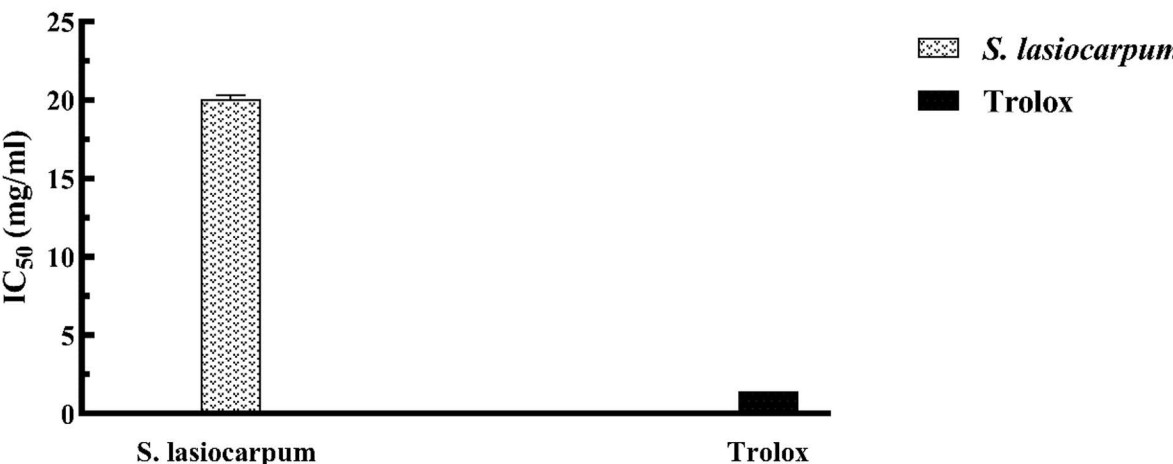

**Fig 1. IC$_{50}$ of of *S. lasiocarpum* in DPPH free radical scavenging activity.** Values are mean ± standard deviation of triplicate determination.

### The activity of α-glucosidase and α-amylase inhibition

The investigation evaluated the inhibitory activities of *S. lasiocarpum* extract and acarbose (used as a control) on α-glucosidase and α-amylase based on their IC$_{50}$ values. The same methods were employed for both the extract and the control. The findings of *S. lasiocarpum* inhibition tests for α-glucosidase and α-amylase are shown in Table 2. According to the findings, the *S. lasiocarpum* extract exhibited the most potent inhibitory action against the two enzymes. For α-glucosidase and α-amylase, the IC$_{50}$ values were 6.31 ± 0.09 mg/mL and 14.06 ± 0.21 mg/mL, respectively.

### Activities of the Dipeptidyl peptidase-4 (DPP-4) enzyme inhibitor

*S. lasiocarpum* extract has inhibitory effects on the DPP-4 enzyme, as shown in Fig 2. Interestingly, the inhibitory effect of *S. lasiocarpum* exceeded that of 5 μg/mL insulin (50.06 ± 0.24%) and 18 nM Sitagliptin (58.14 ± 0.18%) within a specific concentration range. Moreover, the inhibition was directly correlated with the *S. lasiocarpum* concentration. These findings imply that the *S. lasiocarpum* extract greatly influences the anti-DPP-4 enzyme activity.

### Cytotoxic activity on L6 cell lines

The SRB assay results in Fig 3 indicate that the crude extract of *S. lasiocarpum* did not affect cell viability at the specified concentration. However, there was a slight impact on cell viability, with a gradual decrease in the viability of L6 cells as concentrations increased. The IC$_{50}$ value for *S. lasiocarpum* was determined to be 5.02 ± 0.05 mg/ml.

**Table 2. The α-glucosidase and α-amylase inhibitory activities.**

| Sample | α-Glucosidase inhibition, IC50 (mg/mL) | α-Amylase inhibition, IC50 (mg/mL) |
|---|---|---|
| *S. lasiocarpum* | 6.31 ± 0.09 cd | 14.06 ± 0.21 cd |
| Acarbose | 0.0113 ± 0.009[e] | 0.0236 ± 0.023[f] |

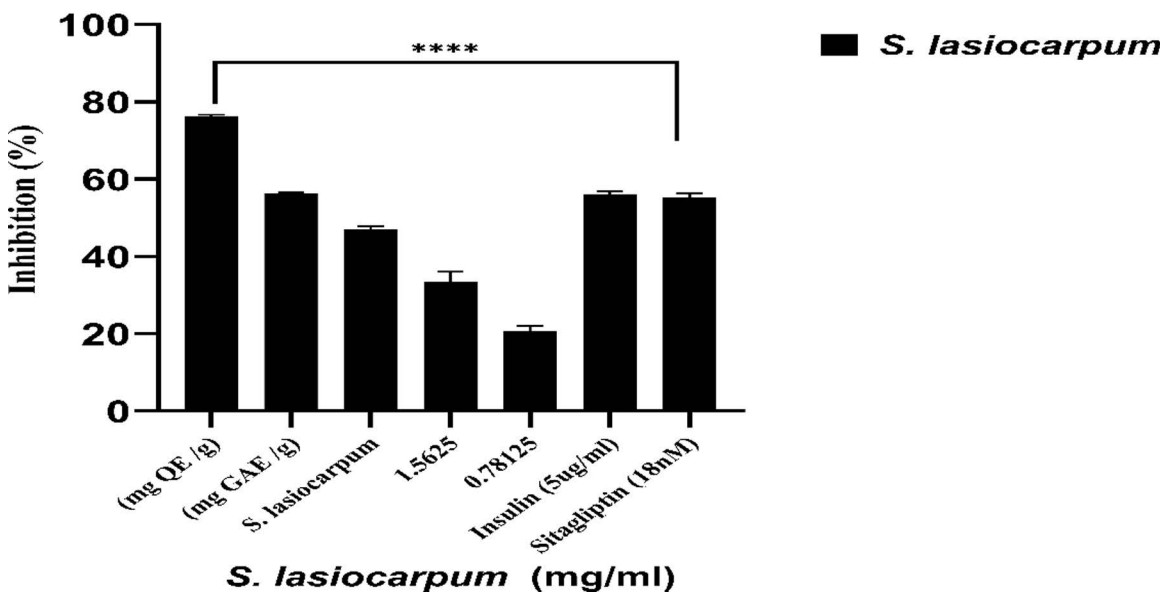

**Fig 2. Inhibitory activity of DPP-4 enzyme.** Values are mean ± standard deviation of triplicate determination. The difference * shows a significant ($p < 0.05$) difference between extract concentrations and Sitagliptin (18nM), *ns: not significant, * < 0.05, ** < 0.01, *** < 0.005, **** < 0.0001*.

These findings suggest that *S. lasiocarpum* has a minimal effect on cell viability. It can be inferred that, under the same concentration conditions, *S. lasiocarpum* inhibits cell proliferation to a limited extent.

## 2-NBDG glucose uptake assay on L6 cell lines

The 2-NBDG glucose kit investigated glucose uptake in L6 cell lines. *S. lasiocarpum* was chosen for the glucose absorption assessment at doses of 125 μg/mL, 62.5 μg/mL, and 31.25 μg/mL (all of which maintained > 70% cell viability) based on cell viability detection. Cells treated with 50 mM Apigenin served as the negative control, while untreated and insulin-treated cells comprised the positive controls.

The results showed that, at different doses, *S. lasiocarpum* extract might improve L6 myoblasts' absorption of glucose ([Fig 4]). In L6 cells, treatment with *S. lasiocarpum* boosted glucose absorption considerably ($p < 0.05$). *S. lasiocarpum* showed 118.19% glucose absorption rates at 31.25 μg/mL compared to untreated cells. Significantly, *S. lasiocarpum* demonstrated the maximum ability to absorb glucose, measuring 62.5 μg/mL and roughly 132.02%.

## Kinetic study of 2-NBDG uptake

Upon investigating the glucose uptake potential, it was essential to identify the specific time point at which *S. lasiocarpum* exhibits maximum glucose uptake and the duration

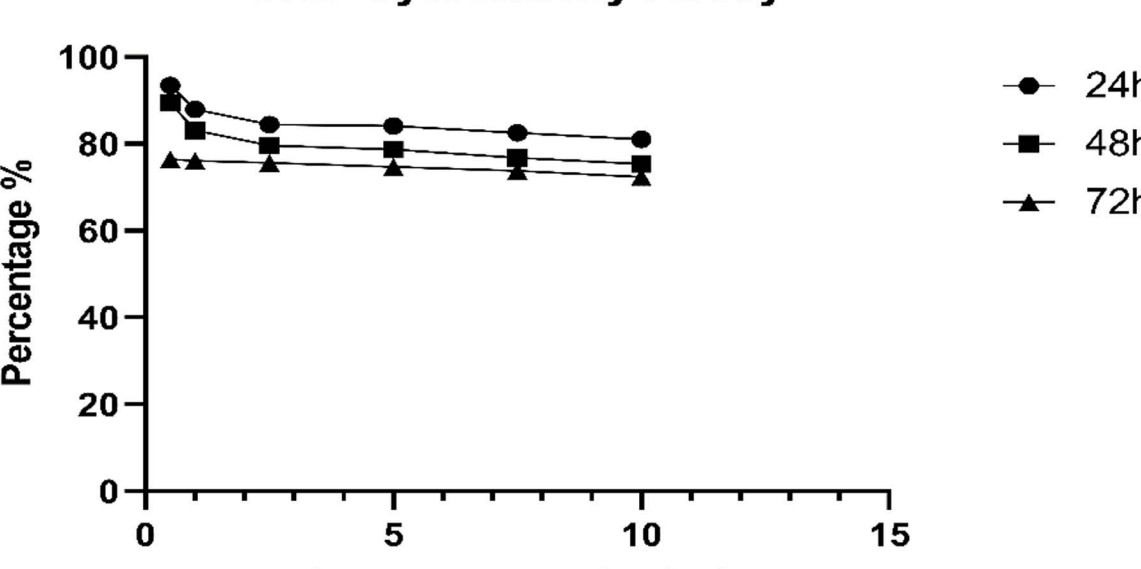

**Fig 3. The effects of the sample on cell viability following a 24-hour treatment period.** The values represent the triplicate determination's mean ± standard deviation. The difference * indicates a significant (p < 0.05) difference between cells that were not treated and cells that were treated with varying doses of *S. lasiocarpum*. ns: *< 0.05, **< 0.01, ***< 0.005, ****< 0.0001; not significant.

needed to activate the glucose uptake pathway for absorbing external glucose into the cell. The study used 2-NBDG, a fluorescent glucose analogue, to assess its uptake rate in L6 rat skeletal muscle cells. 2-NBDG is known to be transported intracellularly by the same glucose transporters (GLUT4) as glucose. The fluorescence density of 2-NBDG, indicating its concentration outside and inside the cells, was measured using a fluorescence spectrophotometer.

In this study, differentiated L6 cells were exposed to the same doses used in the earlier 2-NBDG glucose uptake assay (312.25 μg/mL of *S. lasiocarpum* and one μM of insulin). To obtain kinetic data, wells in 96-well plates containing differentiated L6 cells were divided into different groups and treated with *S. lasiocarpum* and the positive control insulin. The initial group represented 5-minute treatment intervals for up to 30 minutes. Additionally, for each treatment group, an untreated control group and a negative control group were prepared.

The results revealed that insulin-stimulated cells (positive control) showed a significant increase in 2-NBDG uptake within the first 10 minutes, reaching a maximum uptake of 144.181% at the 10-minute mark (Fig 5). The insulin uptake curve decreased sharply and stabilized around 140% for the remaining 30 minutes of stimulation (Fig 5). These findings indicate that *S. lasiocarpum* demonstrates its glucose uptake capacity within 15 minutes, and this potential persists for 30 minutes, albeit at a slightly reduced rate. This highlights its potency as crucial for treating Type 2 Diabetes Mellitus (T2DM).

## Identification of bioactive constituents by HPLC/ LC-MS

Using HPLC/LC-MS, the bioactive components in the crude extract of *S. lasiocarpum* were identified. The bioactive chemicals were first identified by preliminary database annotation and subsequently utilized for in silico research.

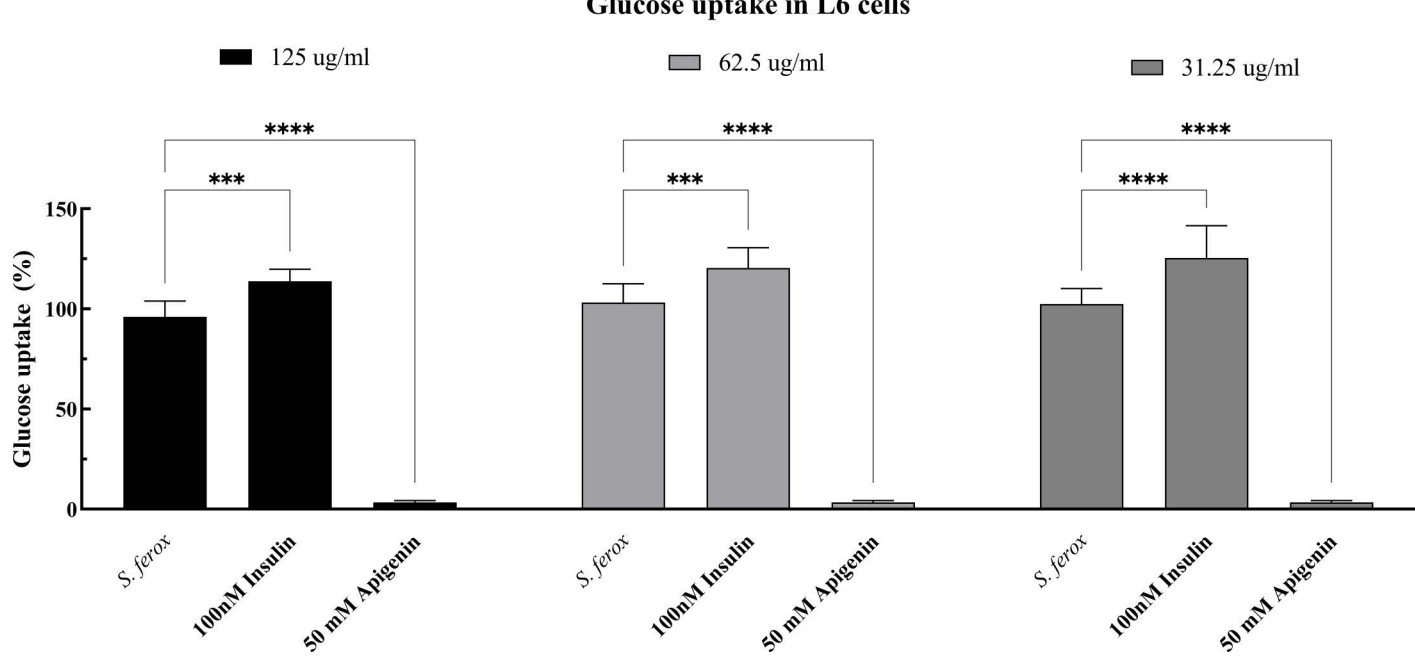

**Fig 4. S. lasiocarpum effect on the absorption of glucose.** The values represent the triple determination's mean ± standard deviation. A significant (p < 0.05) difference between treated and untreated cells is indicated by the difference *. *ns: *< 0.05, **< 0.01, ***< 0.005, ****< 0.0001*; not significant.

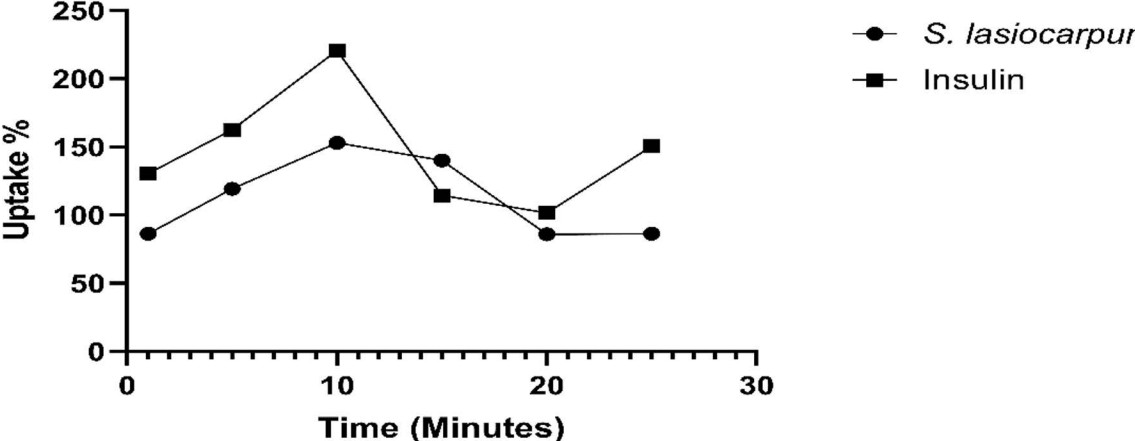

**Fig 5. The kinetic study results show the % of 2-NBDG uptake by differentiated L6 myotubes treated with insulin (positive control) and combination for 30 minutes with 5-minute intervals.** Insulin-stimulated cells showed 2-NBDG uptake in the first 10 minutes, whereas, *S. lasiocarpum* -stimulated cells displayed within 10 minutes. Data are presented as mean ± SEM of at least three independent experiments.

### *In silico* studies

We utilised molecular docking and dynamic simulation techniques to delve deeper into its potential as an anti-diabetic agent, targeting human salivary α-amylase (3DHP) and DPP-4 (6B1E Chain A) proteins.

**Molecular XP docking and MM-GBSA calculation.** Extra-precision (XP) docking, sometimes called flexible docking, is a sophisticated computational approach designed to perform accurate molecular docking calculations directed towards specific objectives. Even though XP docking takes much time, the results are more precise and dependable [31]. An XP score of less than -6 often indicates a stable interaction between a ligand and a ligand-protein complex. After XP docking, 396 and 556 compounds were found to establish moderately stable bindings with the DPP-4 protein (6B1E Chain A) and the α-amylase protein (3DHP), respectively, using a screening threshold of XP Gscore ≤ −6. We plan to analyze these chemicals using MM-GBSA in the future.

Table A.1 contains the results of the MM-GBSA analysis and XP docking as additional materials. Based on the XP docking results, compounds having an XP Gscore < −6 were chosen for MM-GBSA analysis. An MM-GBSA dG Bind indicates a stable binding between the ligand and the protein value less than −30 kcal/mol. Ten compounds satisfy the requirements for the first target protein, according to the MM-GBSA analysis, whereas twenty-one compounds fit the requirements for the second target protein. Table 3 displays the XP Gscore and MM-GBSA binding free energy values for various ligand compounds.

**Determination of Lipinski parameters and ADMET analysis *in silico*.** Our methodology applies ADMET analysis (mostly on adherence to Lipinski's five principles) to ligand molecules discovered by MM-GBSA and XP GScore using Schrodinger's QikProp module. Table 3 elaborates on the compounds that meet Lipinski's five conditions and the qualities that accompany them. Table A.2 is a supplementary file with a detailed list of all the compounds. Out of the total, ten molecules fully comply with Lipinski's five rules. Examining logS and QPP Caco values enables us to anticipate enhanced water solubility and intestinal cell permeability for these compounds. Negative QPlogBB values signify the compounds' polarity and limited blood-brain barrier permeability, with notably low permeability values suggesting minimal central nervous system toxicity risk. Remarkably, the oral absorption rate for these molecules exceeds 50%, indicating a feasible oral administration route. Overall, these compounds exhibit favourable ADMET properties consistent with Lipinski's five rules, rendering them promising candidates for potential anti-diabetic applications. Fig 6 illustrates the interaction of three compounds (selected from the total of ten compounds meeting Lipinski's five rules based on the best XP Gscore and MM-GBSA binding free energy) with the α-amylase (3DHP) protein and DPP-4 (6B1E Chain A) protein through XP docking (3D).

**Molecular dynamics simulations.** The Desmond program refined the ligand-protein complex's binding pattern throughout a 100 ns unrestricted simulation. Maestro 13.5 was also used to create animated trajectory representations and explore interactions. Solvent-accessible

**Table 3. ADMET properties and physicochemical properties using Lipinski's five rules.**

| Compd. No | MW | SASA | QPlogPo/w | QPlogS | QPlogHERG | QPPCaco | QPlogBB | QPlogKhsa | % HOA | No. of violation |
|---|---|---|---|---|---|---|---|---|---|---|
| 1 | 342.388 | 630.385 | 1.299 | −0.101 | −4.288 | 129.082 | −1.947 | −0.843 | 64.134 | 0 |
| 2 | 307.475 | 667.375 | 2.996 | −5.801 | 178.851 | −1.165 | −1.545 | −0.065 | 84.884 | 0 |
| 3 | 205.210 | 381.317 | 1.606 | −1.892 | −3.258 | 32.630 | −0.840 | −0.783 | 42.959 | 0 |
| 4 | 291.476 | 527.723 | 2.837 | −3.798 | 493.454 | 0.241 | 0.489 | −0.857 | 91.758 | 0 |
| 5 | 430.673 | 712.858 | 3.869 | −5.603 | 71.789 | 0.306 | 1.197 | 0.154 | 82.821 | 0 |
| 6 | 183.250 | 372.242 | 3.250 | 428.444 | 1.044 | −0.087 | −4.049 | 848.346 | 85.473 | 0 |
| 7 | 286.336 | 541.212 | 2.305 | −1.082 | −2.967 | 45.367 | −1.388 | −0.193 | 74.586 | 0 |
| 8 | 193.245 | 406.531 | 1.851 | 0.859 | −1.127 | −3.790 | 266.14 | 0.004 | 75.381 | 0 |
| 9 | 332.263 | 552.973 | −2.497 | −1.660 | −4.503 | 4.330 | −3.372 | −3.372 | 10.758 | 0 |
| 10 | 490.576 | 857.689 | 3.768 | −6.424 | −7.449 | 61.223 | −2.796 | −2.796 | 80.993 | 0 |

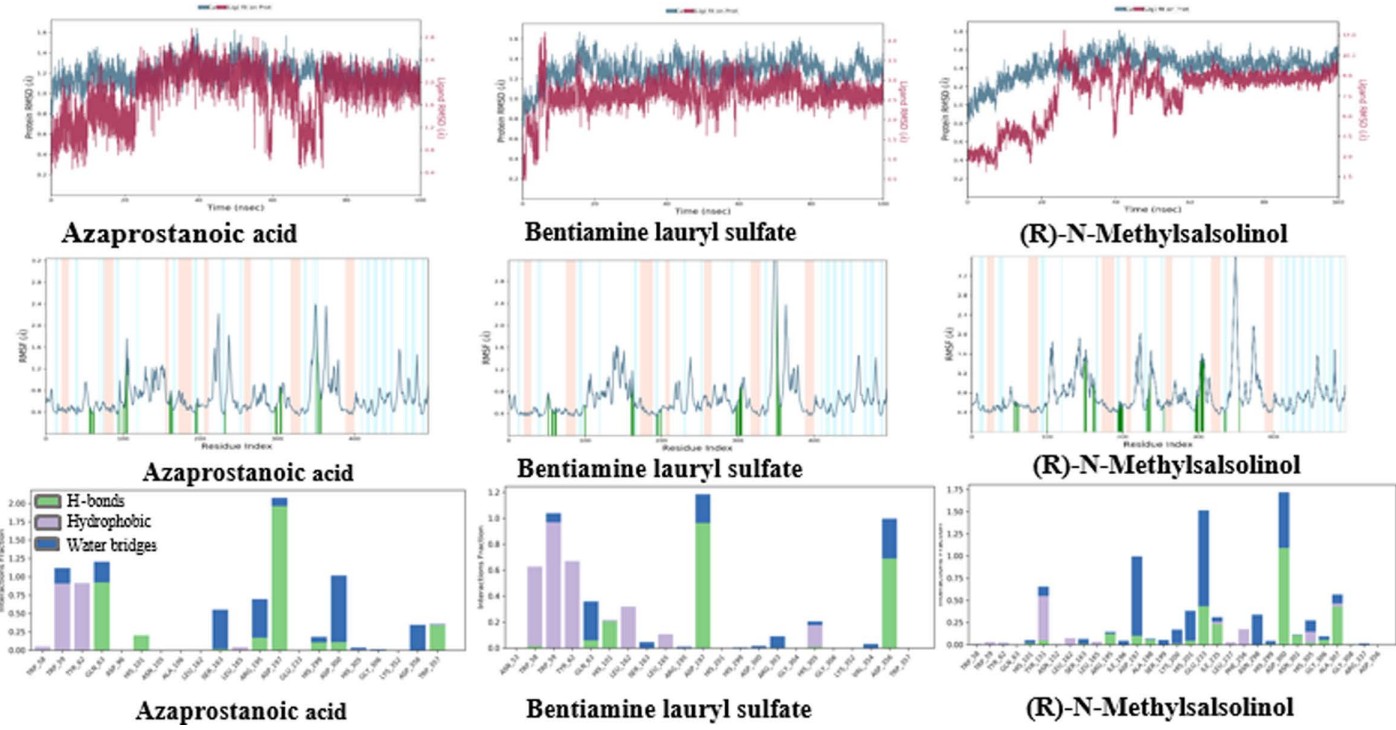

**Fig 6.  The RMSD (a), RMSF (b) trajectories and interaction (c) of DPP-4 (6B1EChain A) protein-ligand (ligands: Azaprostanoic acid, Bentiamine lauryl sulfate, and (R)-N-Methylsalsolinol).**

surface area (SASA), molecular surface area (MolSA), radius of gyration (rGyr), root mean square deviation (RMSD), root mean square fluctuation (RMSF), and intramolecular hydrogen bonds (introHB) were among the plots used to assess the results of the MD simulations, as shown in Fig 7.

**MD Simulation of ligand-α-amylase (3DHP) protein and ligand–DPP-4 (6B1E Chain A) protein.**  This study analyzed the molecular dynamics trajectories of Azaprostanoic acid, Bentiamine lauryl sulfate, and (R)-N-Methylsalsolinol with the DPP-4 protein (6B1E Chain A) using a 100 ns molecular dynamics simulation. Throughout the simulation, we observed and scrutinized the interactions between proteins and ligands, broadly categorized into hydrogen bonding, hydrophobic interaction, ionic interaction, and water bridge all information in Supporting information Table 1A. Fig 8 illustrates that O-acetylcholine binds to the binding pocket surface of the α-amylase protein, engaging in hydrophobic interactions with specific residues such as TRP59, TYR62, ILE235, ALA198, and ILE162, among others. The ligand forms a solitary hydrogen bond with residues ARG195, ASP197, and GLH233, two hydrogen bonds, a salt bridge with residue ASP300, and a π-Cation bond with residue TYR62.

Azaprostanoic acid also attaches to the surface of the α-amylase protein's binding pocket, with hydrophobic pressures exerted by amino acid residues such as LEU162, LEU165, TYR62, and TRP59. The ligand establishes a single hydrogen bond with residues GLY306, ASP300, ARG195, ASP197, and GLH233. Similarly, ®-N-Methylsasolinol interacts with the binding pocket of the α-amylase protein, facilitated by hydrophobic forces from residues like LEU165, ALA198, and ILE235. The ligand forms a single hydrogen bond with GLN63, GLH233, ASP300, HIE299, and ASP197 residues. Conformational strain analyses assessed the ligands' adjustments in maintaining protein-binding complexes.

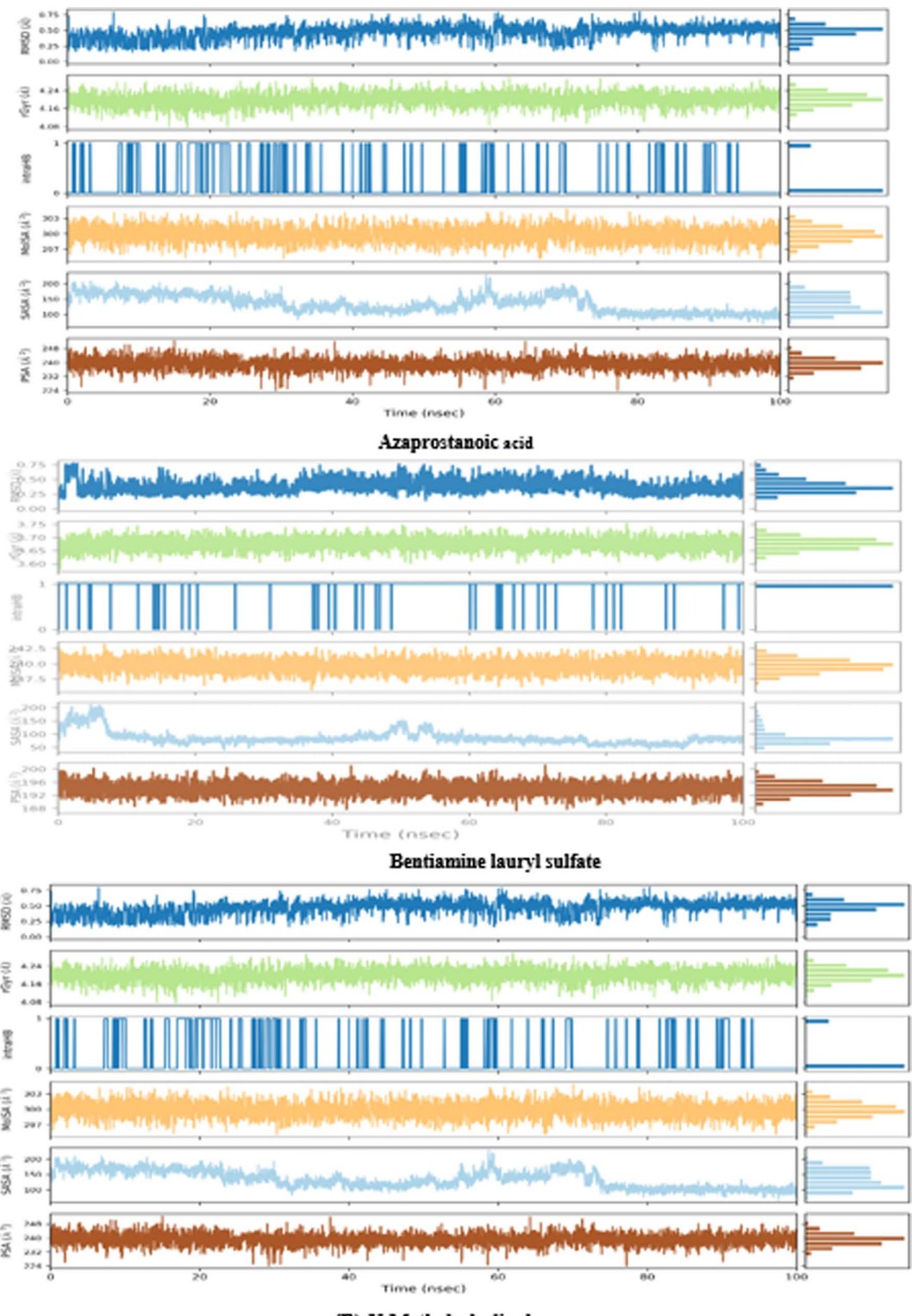

**Fig 7. Variation in the ligands (ligands: Azaprostanoic acid, Bentiamine lauryl sulfate, and (R)-N-Methylsalsolinol) properties w.r.t time during the 100 ns simulation.**

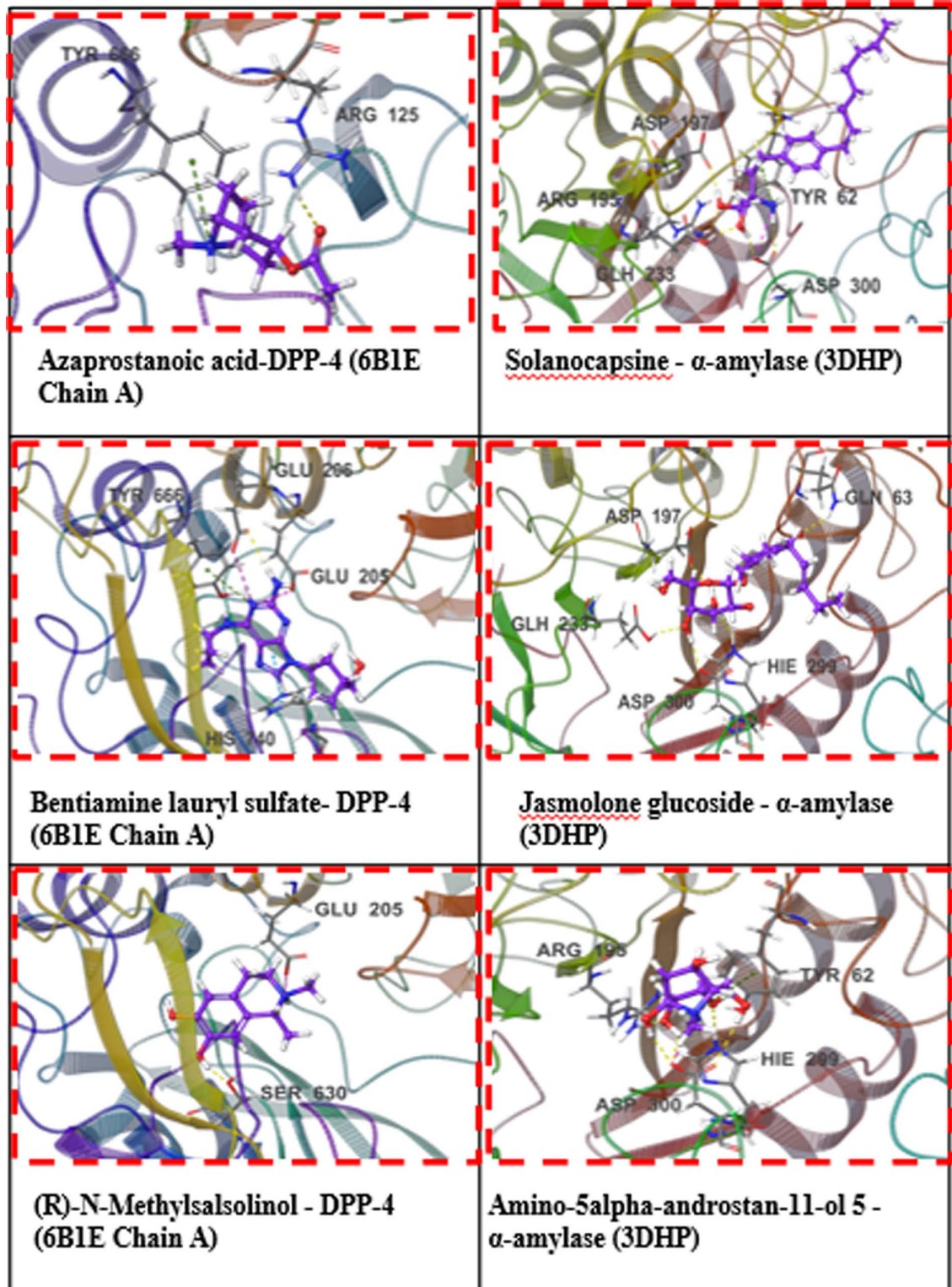

**Fig 8. 3D docking interaction of 3 compounds with α-amylase (3DHP) protein and DPP-4 (6B1E Chain A) protein.**

The α-amylase (3DHP) protein's interactions with ligands Amino-5alpha-androstan-11-ol 5, Jasmolone glucoside, and Solanocapsine over 100 nanoseconds were examined. The RMSD results depicted in Fig 8 demonstrated relative stability between Amino-5alpha-androstan-11-ol 5, Jasmolone glucoside, and Solanocapsine with the DPP-4 protein (Chain A of 6B1E).

Fig 8 further highlights specific residues, such as PHE357, ARG358, TYR547, and GLN553, crucial in the binding interaction between Amino-5alpha-androstan-11-ol 5 and the DPP-4 protein. The critical amino acid residues involved in the Jasmolone glucoside-DPP-4 protein complex include TYR547, CYS551, GLN553, and LYS554. Likewise, in the interaction between Solanocapsine and the DPP-4 protein, the essential residues identified are GLU206, GLN553, LYS554, and TYR666. These interactions predominantly comprise hydrogen bonding, water bridges, and hydrophobic interactions.

## Discussion

Plant extracts are renowned for their rich composition of non-food substances, commonly classified as phytochemicals or nutraceuticals, recognized for their potential to confer various health benefits. Phenolic compounds are a unique class of phytochemicals identified as the main drivers of antioxidant activity. They provide many benefits, such as antibacterial, anti-inflammatory, and antioxidant qualities [32]. These materials are widely used in the food, pharmaceutical, and health products sectors. Within this category, flavonoids, recognized as secondary plant metabolites with various biological actions in vivo and in vitro, especially their antioxidant qualities, represent a significant subgroup [33]. According to Alwasel and Gulcin [30], the DPPH radical scavenging activity method is widely used to evaluate antioxidant activity in food and plant extracts because of its simplicity, sensitivity, speed, reproducibility, and usability. The significant phenolic content in *S. lasiocarpum* fruit extract underscores its potential as a rich source of natural antioxidants, aligning with the study's objective of exploring antidiabetic bioactive compounds. The extract's high DPPH radical scavenging activity reflects its ability to neutralize free radicals, which is crucial for mitigating oxidative stress associated with diabetes [34]. Additionally, the extract's selective cytotoxicity towards certain cancer cell lines suggests its potential for developing complementary therapies for managing diabetes-related complications [35]. These findings collectively support the therapeutic promise of *S. lasiocarpum*, highlighting its multifaceted biological activities in line with the study's aims [36]. Enzymes such as α-glucosidase and α-amylase play pivotal roles in dietary carbohydrate metabolism. Competitive inhibitors targeting these enzymes effectively regulate blood sugar levels by impeding carbohydrate digestion and slowing glucose absorption, presenting practical approaches to managing T2DM [37]. In this study, conducting in-vitro experiments before in-silico simulations helps validate hypotheses, yield tangible results, and provide a realistic environment to study biological interactions that in-silico models may not fully capture. Furthermore, in-vitro experiments offer insights into biological mechanisms such as active-compound interactions. In this study, our approach of prioritizing in-vitro experiments before in-silico simulations presents distinct advantages over traditional methods that often rely on computational predictions alone. While conventional methods may overlook the complexity of biological systems and lead to less accurate predictions, our approach uses real-world data from in-vitro experiments to validate and refine computational models. By grounding in-silico simulations in empirical evidence, we enhance prediction accuracy and gain a more realistic understanding of compound interactions. This iterative process allows insights from *in-vitro* experiments to directly inform and improve computational simulations, addressing discrepancies between predictions and biological responses and ultimately providing a more reliable evaluation of compound efficacy and mechanisms.

Serine protease Dipeptidyl peptidase-4 (DPP4) is an important therapeutic target for managing diabetes since it is essential to metabolising insulin and glucose [38]. Individual and combination extracts were assessed using the fluorescence method for in vitro inhibitory activities against DPP-4. Additionally, the cytotoxicity of six crude extracts was evaluated using L6 myoblast cells treated with various doses for 24 hours, employing the sulphorhodamine B (SRB) assay to observe cytotoxic effects on differentiated L6 cells. [39].

*S. lasiocarpum* demonstrated notable potential for glucose uptake, suggesting the presence of insulin-mimetic compounds. Glucose uptake in skeletal muscle cells is a crucial and rate-limiting step for its utilization [40]. Molecular docking and dynamic modelling experiments explored potential anti-diabetic characteristics, concentrating on the target proteins DPP-4 (6B1E Chain A) and human salivary α-amylase (3DHP). Although it takes much time, the Extra-Precision (XP) docking method, sometimes called flexible docking, is a sophisticated computational technique for high-resolution molecular docking simulations suited to specific targets. The results it produces are more accurate and reliable [31]. Prioritizing the analysis of absorption, distribution, metabolism, excretion, and toxicity (ADMET) features is necessary to assess the bioavailability and toxicity of investigated substances [41]. According to Gajjar et al. [42], one of the most important computational methods for determining the ligands' time-dependent stability in the receptor complex's active region is molecular dynamics (MD) simulation. We incorporated well-established reference compounds to validate our computational findings and benchmark the efficacy of the new compounds. Sitagliptin, a recognized DPP-4 inhibitor, was chosen to compare our compounds' binding affinities and inhibitory potential against DPP-4 [43]. For human salivary α-amylase inhibition, Acarbose, a known competitive inhibitor, was used as a benchmark [44,45]. Using these reference compounds is essential for validating the accuracy and relevance of our in-silico models, contextualizing the performance of new compounds, and highlighting any potential advantages over existing treatments. Additionally, incorporating reference drugs is crucial for assessing the bioavailability and toxicity profiles of the new compounds through ADMET analysis [41].

These findings suggest that *S. lasiocarpum* could offer therapeutic benefits beyond diabetes management, potentially addressing cancer-related complications. Future research should focus on isolating specific phenolic compounds better to understand their contributions to antioxidant and cytotoxic activities. Additionally, exploring in vivo models and clinical trials could further validate the efficacy and safety of *S. lasiocarpum* for broader therapeutic applications, including its role in managing diabetes and related conditions.

## Conclusions

The current study explored the possibility of *S. lasiocarpum* in treating diabetes using laboratory experiments and computer simulations. The *in vitro* assessments showed that *S. lasiocarpum* had a low antioxidant activity compared to trolox, as demonstrated by its ability to scavenge DPPH radicals. In addition, the *S. lasiocarpum* exhibited increased enzyme inhibitory activity against α-amylase, α-glucosidase, and DPP-4. Cell investigations confirmed these results, showing a statistically significant ($p < 0.05$) rise in glucose absorption by L6 myoblasts with each extract and combination. The *S. lasiocarpum* has a substantial impact on increasing glucose uptake.

The effectiveness of the polyherbal combinations in treating diabetes was further evaluated utilizing computational research approaches. The compounds discovered through LC-MS analysis were subjected to molecular docking, MM-GBSA calculation, Lipinski parameter evaluation, ADMET analysis, and MD simulation, targeting proteins α-amylase and DPP-IV. The results indicate that the *S. lasiocarpum* contains components that have inhibitory effects on α-amylase and DPP-4 enzymes. These chemicals are assumed to play a role in the observed

enzyme inhibitory activity in laboratory settings. The novelty of our approach lies in conducting in-vitro experiments before in-silico simulations, enhancing model accuracy by grounding computational predictions in empirical data. This iterative process refines simulations with real-world insights, provides a comprehensive understanding of compound interactions, and identifies discrepancies early. This method ensures more reliable predictions and a more efficient drug development pipeline, offering a more accurate and practical approach than traditional methodologies. Our research findings from laboratory experiments and computer simulations suggest that *S. lasiocarpum* has beneficial effects in treating diabetes. Consequently, it can be considered a viable supplementary formulation for preventing and treating diabetes.

## Supporting information

**S1A Table. The results of the MM-GBSA analysis and XP docking as additional materials. Based on the XP docking results, compounds having an XP Gscore < −6 were chosen for MM-GBSA analysis.**
(XLSX)

## Author contributions

**Conceptualization:** Jing Zhao.

**Data curation:** Jing Zhao, Wamidh H. Talib.

**Formal analysis:** Ahmed Abdulkareem Najm.

**Investigation:** Jing Zhao.

**Methodology:** Ahmed Abdulkareem Najm.

**Resources:** Jing Zhao, Partha Pratim Dutta.

**Software:** Jing Zhao, Zhang Yu Ming.

**Supervision:** Douglas Law, Shazrul Fazry.

**Validation:** Ahmed Abdulkareem Najm, Ibrahim Mahmood, Zhang Yu Ming, Partha Pratim Dutta.

**Visualization:** Jing Zhao, Ibrahim Mahmood.

**Writing – original draft:** Ahmed Abdulkareem Najm.

**Writing – review & editing:** Ahmed Abdulkareem Najm, Douglas Law, Shazrul Fazry.

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
