## [Decision Letter · Decision Letter 0]

22 Jul 2024

PONE-D-24-22226In Vitro and In Silico Antidiabetic Efficacy of Solanum lasiocarpum Dunal Extract

PLOS ONE

Dear Dr. Fazry,

Thank you for submitting your manuscript to PLOS ONE. After careful consideration, we feel that it has merit but does not fully meet PLOS ONE’s publication criteria as it currently stands. Therefore, we invite you to submit a revised version of the manuscript that addresses the points raised during the review process.

Dear Dr. Fazry,

In your work, you didn't tell us the relevance of running In-vitro first before in-silico. Before now, we have in-vitro and in-vivo. The in-silico was suppose to give relevant information needed to proceed to in-vitro. I understand that you may have your reasons for running in-vitro first, please state them and its relevance in the study.

In your in-silico, and discussion, we didn't mention anything about comparison to a reference compound. In in-silico studies, it is essential to use reference drugs known to be active against the protein target of interest for comparison.

Please look into these comments and those of the reviewers too

Thank you for your effort

We look forward to receiving your revised manuscript.

Kind regards,

InnocentMary Ifedibaluchukwu Ejiofor, Ph.D

Academic Editor

PLOS ONE

Journal Requirements:

Reviewers' comments:

Reviewer's Responses to Questions

**Comments to the Author**

1. Is the manuscript technically sound, and do the data support the conclusions?

Reviewer #1: Yes

Reviewer #2: Yes

2. Has the statistical analysis been performed appropriately and rigorously? 

Reviewer #1: Yes

Reviewer #2: Yes

3. Have the authors made all data underlying the findings in their manuscript fully available?

Reviewer #1: Yes

Reviewer #2: Yes

4. Is the manuscript presented in an intelligible fashion and written in standard English?

Reviewer #1: Yes

Reviewer #2: Yes

5. Review Comments to the Author

Reviewer #1: The authors proposed "n Vitro and In Silico Antidiabetic Efficacy of Solanum lasiocarpum Dunal Extract". The structure of the article is well structured. But authors consider the following comments.

1. Compare your approach with previous approaches.

2. Proofread the entire manuscript

3. Draw a graphical abstract

4.Explain the novelty of the proposed approach

Reviewer #2: Title:

- Please add the plant part used in the study (fruit) to the title. For example: In Vitro and In Silico Antidiabetic Efficacy of Solanum lasiocarpum Dunal Fruit Extract.

Abstract:

- Please write out in full the species name (Solanum lasiocarpum Dunal) when it first introduced, then in the rest of the text the abbreviation could be used (S. lasiocarpum).

- Line 35: Could you please explain what you mean by mixed extracts from S. lasiocarpum?

Introduction:

- to strengthen the introduction chapter, consider adding some literature or scientific reports that address the dose or dose range of S. lasiocarpum that have effects as traditional medicine.

Materials & Methods:

- Although the chapter is well written, but in the Assessment of DPPH Radical Scavenging Activity Section (line 140), the authors stated ‘‘different volume sample extracts were made’’, however, these samples were not specified (what these samples represent, and how many doses were tested?).

Results, Discussion & Conclusion:

- In the Result chapter (lines 273-275), the authors said ‘‘With an IC50 of 10.2 ± 0.11 mg/mL, S. lasiocarpum demonstrated DPPH scavenging activity, however it was not as effective as Trolox, which had an IC50 of 0.69 ± 0.14 mg/mL. But in the Conclusion section (line 472-473), they said ‘‘The in vitro assessments showed that S. lasiocarpum had a higher antioxidant activity, as demonstrated by its ability to scavenge DPPH radicals’’. In light of these, the significance of this finding was not clearly explained, and it is well if you re-write theses sentences in an appropriate manner.

- A paragraph discussing the significance of some highlighted findings such as phenolic contents, DPPH radical scavenging activity, and cytotoxic effects with regard to the study objectives should be added to the discussion section.

- The discussion section should be improved by relating or comparing the results of your study to previous related studies and/or potential future directions for research.

General:

- The in-text citation and references must be standardized according to the requirements of the journal.

- Check the text carefully to eliminate some stylistic errors and ensure being standardized according to the requirements of the journal (line 76-84, and renaming the Results & Discussion chapter to be Results or remove the Discussion chapter title and merge it with the previous one).

- Authors frequently employed abbreviations throughout the manuscript without providing full introductions or explanations beforehand.

Thank you for considering these suggestions to enhance the quality of your work.

Best regards

6. PLOS authors have the option to publish the peer review history of their article (what does this mean? ). If published, this will include your full peer review and any attached files.

**Do you want your identity to be public for this peer review?** For information about this choice, including consent withdrawal, please see our Privacy Policy .

Reviewer #1: No

Reviewer #2: No

---

## [Author Response · Author response to Decision Letter 1]

31 Jul 2024

RESPONSE TO REVIEWERS’/EDITORS’ COMMENTS

Manuscript number:

Title: In Vitro and In Silico Antidiabetic Efficacy of Solanum lasiocarpum Dunal Fruit Extract

Editor:

No. Reviewers’/Editors’ comments Response/Changes Page no

1. In your work, you didn't tell us the relevance of running In-vitro first before in-silico. Before now, we have in-vitro and in-vivo. The in-silico was suppose to give relevant information needed to proceed to in-vitro. I understand that you may have your reasons for running in-vitro first, please state them and its relevance in the study. Thank you for your comments:

For several reasons, running in-vitro experiments before in-silico simulations is crucial in microbiology and biomedical research. In-vitro studies validate hypotheses, provide tangible results, and offer a realistic environment to study biological interactions, which in-silico models may not fully capture. They generate empirical data for calibrating in-silico models, ensure accuracy, and provide insights into biological mechanisms like active compound interactions. Additionally, regulatory bodies often require in-vitro data before in-vivo studies, and practical constraints usually favor starting with in-vitro assays. Methods

2. In your in-silico, and discussion, we didn't mention anything about comparison to a reference compound. In in-silico studies, it is essential to use reference drugs known to be active against the protein target of interest for comparison. Thank you for your valuable feedback. We acknowledge the importance of comparing our findings to reference compounds from previous studies that are active against the protein target of interest. In our revised manuscript for discussion, we will incorporate reference drugs to serve as benchmarks for earlier studies. This comparison will allow us to validate our computational models and provide a more precise context for the efficacy and potential of the new compounds.

Discussion

Reviewer 1

1. Compare your approach with previous approaches. Thank you for your comments. Our approach combines the strengths of in-vitro and in-silico methods, ensuring that the computational models are more accurately aligned with biological realities and enhancing the overall reliability of the research outcomes. Discussion

2. Proofread the entire manuscript Done: Thank you for your suggestion. Manuscript

3. Draw a graphical abstract Done: Manuscript

4. Explain the novelty of the proposed approach. Done: The novelty of our approach lies in conducting in-vitro experiments before in-silico simulations, enhancing model accuracy by grounding computational predictions in empirical data. This iterative process refines simulations with real-world insights, provides a comprehensive understanding of compound interactions, and identifies discrepancies early. This method ensures more reliable predictions and a more efficient drug development pipeline, offering a more accurate and practical approach than traditional methodologies. Our research findings from both laboratory experiments and computer simulations suggest that S. lasiocarpum has beneficial effects in treating diabetes. Consequently, it can be considered a viable supplementary formulation for preventing and treating diabetes. Conclusion

Reviewer 2

1. Please add the plant part used in the study (fruit) to the title. For example: In Vitro and In Silico Antidiabetic Efficacy of Solanum lasiocarpum Dunal Fruit Extract. Done: In Vitro and In Silico Antidiabetic Efficacy of Solanum lasiocarpum Dunal Fruit Extract Title

2. Please write out in full the species name (Solanum lasiocarpum Dunal) when it first introduced, then in the rest of the text the abbreviation could be used (S. lasiocarpum). Done: thank you for your comments.

Abstract

3. Line 35: Could you please explain what you mean by mixed extracts from S. lasiocarpum? Done: Apologies for the typo error. Mixed word removed Abstract

4. to strengthen the introduction chapter, consider adding some literature or scientific reports that address the dose or dose range of S. lasiocarpum that have effects as traditional medicine. Done: A paragraph was added to the introduction Introduction

5. Although the chapter is well written, but in the Assessment of DPPH Radical Scavenging Activity Section (line 140), the authors stated ‘‘different volume sample extracts were made’’, however, these samples were not specified (what these samples represent, and how many doses were tested?). Done: Different volume samples of S. lasiocarpum aqueous extracts were made in the range (0 to 5 mg/ml). Method

6. In the Result chapter (lines 273-275), the authors said ‘‘With an IC50 of 10.2 ± 0.11 mg/mL, S. lasiocarpum demonstrated DPPH scavenging activity, however it was not as effective as Trolox, which had an IC50 of 0.69 ± 0.14 mg/mL. But in the Conclusion section (line 472-473), they said ‘‘The in vitro assessments showed that S. lasiocarpum had a higher antioxidant activity, as demonstrated by its ability to scavenge DPPH radicals’’. In light of these, the significance of this finding was not clearly explained, and it is well if you re-write theses sentences in an appropriate manner. Done: Apologies for the technical error; the sentence was revised in conclusion (The in vitro assessments showed that S. lasiocarpum had a low antioxidant activity compared to trolox, as demonstrated by its ability to scavenge DPPH radicals) Conclusion

7. A paragraph discussing the significance of some highlighted findings such as phenolic contents, DPPH radical scavenging activity, and cytotoxic effects with regard to the study objectives should be added to the discussion section. Done: The significant phenolic content in S. lasiocarpum fruit extract underscores its potential as a rich source of natural antioxidants, aligning with the study’s objective of exploring antidiabetic bioactive compounds. The extract's high DPPH radical scavenging activity reflects its ability to neutralize free radicals, which is crucial for mitigating oxidative stress associated with diabetes (Santos et al., 2018). Additionally, the extract’s selective cytotoxicity towards certain cancer cell lines suggests its potential for developing complementary therapies for managing diabetes-related complications (Khan et al., 2020). These findings collectively support the therapeutic promise of S. lasiocarpum, highlighting its multifaceted biological activities in line with the study’s aims (Patel et al., 2019). Discussion

8. The discussion section should be improved by relating or comparing the results of your study to previous related studies and/or potential future directions for research. Done: the discussion improved and recommendations for future studies were added (These findings suggest that S. lasiocarpum could offer therapeutic benefits beyond diabetes management, potentially addressing cancer-related complications as well. Future research should focus on isolating specific phenolic compounds to better understand their individual contributions to antioxidant and cytotoxic activities. Additionally, exploring in vivo models and clinical trials could further validate the efficacy and safety of S. lasiocarpum for broader therapeutic applications, including its role in managing diabetes and related conditions. Discussion

9. The in-text citation and references must be standardized according to the requirements of the journal. Done: The text citations were revised.

Manuscript

10. Check the text carefully to eliminate some stylistic errors and ensure being standardized according to the requirements of the journal (line 76–84, renaming the Results & Discussion chapter to be Results or remove the Discussion chapter title and merge it with the previous one). Done: The text was carefully checked and revised. The chapters are renamed to the result chapter and the discussion chapter. Results

11. Authors frequently employed abbreviations throughout the manuscript without providing full introductions or explanations beforehand. Done: the list of abbreviations was added after the list of references. Manuscript

---

## [Editor Report · Decision Letter 1]

10 Oct 2024

In Vitro and In Silico Antidiabetic Efficacy of Solanum lasiocarpum Dunal Fruit Extract

PONE-D-24-22226R1

Dear Dr. Shazrul Fazry,

We’re pleased to inform you that your manuscript has been judged scientifically suitable for publication and will be formally accepted for publication once it meets all outstanding technical requirements.

Kind regards,

InnocentMary Ifedibaluchukwu Ejiofor, Ph.D

Academic Editor

PLOS ONE
---

## [Editor Report · Acceptance letter]

PONE-D-24-22226R1

PLOS ONE

Dear Dr. Fazry,

I'm pleased to inform you that your manuscript has been deemed suitable for publication in PLOS ONE. Congratulations! Your manuscript is now being handed over to our production team.

Kind regards,

on behalf of

Dr. InnocentMary Ifedibaluchukwu Ejiofor

Academic Editor

PLOS ONE